# Material Properties of HY 80 Steel after 55 Years of Operation for FEM Applications

**DOI:** 10.3390/ma14154213

**Published:** 2021-07-28

**Authors:** Szturomski Bogdan, Kiciński Radosław

**Affiliations:** Mechanical and Electrical Engineering Department, Polish Naval Academy, 81-103 Gdynia, Poland; bsztur@gmail.com

**Keywords:** HY 80 steel, static tensile test, dynamic tensile test, rotary hammer, plastic characteristics, Johnson–Cook material model

## Abstract

The paper presents the results of testing the properties of HY 80 steel from the hull of a Kobben class 207 submarine after 60 years of operation in extreme sea conditions. Steels from the HY family in the post-war period were used to build American and German submarines. For the obtained fragment of steel from the hull of the Polish submarine ORP Jastrząb (ORP-Boat of the Republic of Poland), static tensile tests were performed on an MTS testing machine. Dynamic tensile tests were carried out on a rotary hammer for the strain rate in the range of 500~2000 s^−1^. Results: Based on the obtained results, the Johnson–Cook model and the failure parameters of HY 80 steel in terms of the finite element method (FEM) were developed. Conclusion: This model can be used to simulate fast-changing processes such as resistance of structures to collisions, shelling, and the impact of pressure waves caused by explosions in water and air related to submarines.

## 1. Introduction

Steels of increased strength have been used to construct the hulls of submarines, the structure of which is exposed to high loads caused by water pressure when submerged, impacts when lying on the bottom, and the effects of explosions of underwater mines and depth bombs. Commonly used by the United States for shipbuilding in the post-war years were HY 80, HY 100, HY 130, and HY 200 [1,2]. In 1960, HY 80 steel was used to make the hull of the first nuclear-powered submarine USS Thresher (SSN-593), which unfortunately ended its service tragically in April 1963.

The hulls of Los Angeles type units-USS Providence (SSN-719) [3] were made of HY 80 steel [4]. German shipyards also used this steel for the construction of project 205 submarines and their modifications. Until 2019, there were four Kobben-class (project 207) submarines in operational use by the Polish Navy. These were (S-306 Skolpen) ORP “Sep”, (S-308 Stord) ORP “Sokol”, (S-309 Swenner) ORP “Bielik”, and (S-319 Kunna) ORP “Kondor” (Figure 1). The fifth submarine, (S-318 Kobben) ORP “Jastrzab”, is used as a crew training simulator at the Polish Naval Academy. They are the last ships of this series worldwide. In 1964–1967, 15 such units were built at the German shipyard Rheinstahl Nordseewerke to modify the 205 project submarines for Norway’s Navy. Five of them were transferred to the Polish Navy in 2002–2003 [5]. These ships end their service in the Polish Navy. Due to the decommissioning, their tactical data can be declassified, and therefore, their documentation can be used for scientific purposes.

HY 80 (High Yield 80) steel can withstand a load of 80 pounds per square inch which is equivalent to approximately 551.5 MPa. Their catalogue yield point is 552 MPa [4]. It is an iron alloy with a content of 0.12–0.2% carbon, 2–3.5% nickel with the addition of chromium, molybdenum, and copper. Owing to these additives, they are characterised by increased strength, good plasticity, impact strength, and corrosion resistance. Moreover, steels of this type show good weldability, making it possible to build ships in sections and then join them [2].

In recent years, the number of accidents to submarines made of HY80 steel has increased. As a result of this study, submarine engineers have the opportunity to analyse the strength of submarine hulls. They can decide to allow them for further use. The conducted tests will allow the properties of the steel to be assessed after 55 years of operation to discern whether they have changed, or whether the ship’s hull can be further used.

## 2. Materials and Methods

The adaptation of ORP “Jastrząb” (S-318 Kobben, built: 1966) (Figure 2) as a submarine simulator at the Naval Academy in Gdynia required numerous modifications to the hull, including vents and air conditioning connections. The material that had been removed to make holes for conditioning purposes was obtained as strength test specimens.

The obtained material was used to make standardised samples for the quasi-static tensile test on the MTS testing machine with a diameter of 8 mm in accordance with EN ISO 6892-1: 2016 [7] (Figure 3). For the dynamic tensile test on a rotary hammer, round specimens with a thread with a working part diameter of 5 mm and a length of 40 mm were made. The samples were made along the ship’s axis–axial.

## 3. Results

### 3.1. Uniaxial Static Tensile Test

Samples were prepared from the obtained material, and a static tensile test was performed. The test was carried out on four samples. The test results are summarised in Figure 4.

### 3.2. The Study of Dynamic Mechanical Properties Using a Rotary Hammer

The Fundamentals of Technology Laboratory of the Naval Academy in Gdynia has a unique stand-a rotary hammer (Figure 5) that enables performing the dynamic tensile test at speeds in the range of 10–50 m/s. With a sample length of 20 mm, this allows the strain rate to be equal to 500–2000 s^−1^. The measurement results are presented in Table 1. 

At the rotary hammer laboratory stand, the sample breaking force is recorded at a given strain rate in the range of 0–2000 s^−1^. The maximum breaking force is then converted into the stress corresponding to the ultimate strength in dynamic tensile strength. The strain rate is defined as the ratio of the tearing speed of the sample to its measured length [8] as follows:(1)$$\dot{\varepsilon }=\frac{d\varepsilon }{dt}=\frac{d}{dt}\left(\frac{\upsilon \cdot t}{l}\right)=\frac{\upsilon }{l}$$

### 3.3. HY 80 True Characteristics

The relationship between the true stresses *σ*_true_ and nominal stresses *σ*_nom_ obtained from the tensile test is obtained assuming that the volume of the stretched sample during stretching is constant; thus,
(2)$$l_{0}\cdot A_{0}=l\cdot A\left(F\right)$$

Hence,
(3)$$\sigma_{\mathrm{true}}=\frac{F}{A\left(F\right)}=\frac{F}{A_{0}}\frac{l}{l_{0}}=\sigma_{\mathrm{nom}}\left(\frac{l}{l_{0}}\right)$$

Since
(4)$$\frac{l}{l_{0}}=1+\varepsilon_{\mathrm{nom}}$$
we have, therefore,
(5)$$\varepsilon_{\mathrm{true}}=\ln\left(1+\varepsilon_{\mathrm{nom}}\right).\;$$
(6)$$\sigma_{\mathrm{true}}=\sigma_{\mathrm{nom}}\left(1+\varepsilon_{\mathrm{nom}}\right)$$

Plastic deformation is the difference between the true deformation *ε*_true_, and the elastic deformation *ε*_el_.
(7)$$\varepsilon_{\mathrm{pl}}=\varepsilon_{\mathrm{true}}-\varepsilon_{\mathrm{el}}=\varepsilon_{\mathrm{true}}-\frac{\sigma_{\mathrm{true}}}{E}$$

According to the above formulas, the true and plastic characteristics for the tested HY 80 steel samples were developed (Figure 6). The basic material constants describing the tested steel are summarised in Table 2.

The chemical composition of the breakthrough structure and the material model proposal was also considered in the paper [2]. However, in CAE programs, functions are used to describe the plastic characteristic depending on the strain rate and temperature $\sigma_{\mathrm{true}}=\sigma_{\mathrm{true}}\;\left(\varepsilon_{\mathrm{pl}},\dot{\varepsilon },\;\theta \right)$. In the case of metals, the Johnson–Cook constitutive model has become the most frequently used standard [9]. In this model, the plastic Huber–Mises–Hencky (HMH) reduced stresses σ_pl_ are described by the following equation:(8)$$\sigma_{\mathrm{pl}}=\left(A+B\varepsilon_{\mathrm{pl}}^{n}\right)\left[1+C\ln\left(\frac{\dot{\varepsilon }}{{\dot{\varepsilon }}_{0}}\right)\right]\left[1-{\left(\frac{\theta -\theta_{0}}{\theta_{\mathrm{melt}}-\theta_{0}}\right)}^{m}\right]$$
where
A–elastic range of the material $\sigma_{\mathrm{pl}=0}$ (it is often simplified in form A = *R*_e_);B–hardening parameter;n–hardening exponent;C–strain rate coefficient;$\varepsilon_{\mathrm{pl}}$–true plastic strain;$\dot{\varepsilon }\;$–strain rate;${\dot{\varepsilon }}_{0}$–quasi-static strain rate (0.0001 s^−1^);*θ*–current material temperature;*θ*_0_–ambient temperature;*θ*_tmelt_–melting temperature;m–thermal softening exponent.

The above values for this model are determined based on the static tensile test and Hopkinson or Taylor tests [10]. However, with strain rates ranging up to 2000 s^−1^, these data can be obtained from a rotary hammer tensile test.

The parameters A, B, C, n, and m can be determined in many other ways [11]. One of the ways is the so-called engineering formula, according to which the parameters of the first term A, B, and n are determined based on the results of the static tensile test according to the following algorithm:

*R_m_, ε_m_, E* should be determined from true characteristics, along with the values of the A point corresponding to $\sigma_{\mathrm{pl}=0}$, $\varepsilon_{\mathrm{pl}=0}$, which constitute the elastic range of the material behaviour; then, according to the Formulas (9)–(12), calculate the *R*_e,true_*, R*_m,true_*, ε*_m,true_*, ε*_m,pl_ values as follows:(9)$$A=\sigma_{\mathrm{pl}=0}$$
(10)$$R_{m,\mathrm{true}}=R_{m}\left(1+\varepsilon_{m}\right).\;$$
(11)$$\varepsilon_{m,\mathrm{true}}=\ln\left(1+\varepsilon_{m}\right)$$
(12)$$\varepsilon_{m,\mathrm{pl}}=\varepsilon_{m,\mathrm{true}}-\frac{R_{m,\mathrm{true}}}{E}$$
determine the parameters B, n [11] according to the following Formulas (13) and (14):(13)$$n=\frac{R_{m,\mathrm{true}}\cdot \varepsilon_{m,\mathrm{pl}}}{R_{m,\mathrm{true}}-A}$$
(14)$$B=\frac{R_{m,\mathrm{true}}-A}{\varepsilon_{m,\mathrm{pl}}^{n}}$$

Taking the average values from Table 2 and using the Formulas (9), (13) and (14), the coefficients for the first component of the Johnson–Cook constitutive model were determined, which are the following:
A = 559 MPa;B = 518 MPa;n = 0.379.

Figure 7 shows the compilation of the nominal characteristic from the MTS machine (red), the true characteristic determined from Equations (5) and (6) (blue), and the JC model (the first part of the Equation (8) (green).

To determine the C parameter, it is necessary to know the value $\;R_{m,\;\mathrm{true}}\left(\dot{\varepsilon }\right)\;$ or a given strain rate determined during the dynamic tensile test on a rotary hammer (Table 3). From transforming Equation (8), we obtain
(15)$$C=\left(\frac{R_{m,\mathrm{true}}\left(\dot{\varepsilon }\right)}{R_{m,\mathrm{true}}\left(\dot{\varepsilon_{0}}\right)}-1\right)/\ln\left(\frac{\dot{\varepsilon }}{{\dot{\varepsilon }}_{0}}\right)$$

Based on the calculations, the mean value of the C parameter was determined, C = 0.0268. In Figure 8, the influence of the C parameter on material behaviour is shown. Figure 8 shows the behaviour of the material as a function of strain rate in the Johnson–Cook model (first and second term of Equation (8) against the background of the real characterisation determined from Equation (6).

The values for the temperature component can be taken based on the literature [1,4], and they are similar for most steels; thus,
Ambient temperature θ_0_ = 293.15 K;Melting temperature θ_top_ = 1733 ~ 1793;Thermal coefficient m = 0.75 ÷ 1.15.

### 3.4. HY 80 Steel Failure at Uniaxial Tension

The material failure model used in CAE programs is detailed in several studies [6,8,12,13,14]. The value of the destructive deformation is a function of the so-called stress state indicator *η*_TRIAX_ (stress triaxiality). It is the ratio of the pressure being the mean of the principal stresses to the Huber–Mises–Hencky reduced stress *σ*_HMH_ [12,13]
(16)$$\eta_{\mathrm{TRIAX}}=\frac{p}{\sigma_{\mathrm{HMH}}}$$

In a three-dimensional state of stress, the pressure is
(17)$$p=\frac{1}{3}\left(\sigma_{1}+\sigma_{2}+\sigma_{3}\right).\;$$

For the uniaxial stretching state, the value of the triaxiality coefficient is equal to 0.33 (Table 4).

## 4. Discussion

The failure mechanism for HY 80 steel is shown in the true characteristic diagram *σ_true_-ε _true_* (Figure 9). The elastic range is between points 0 and 1. Between points 1 and 2, there is a plastic range (hardening). In point 2, the destruction process is initiated. After crossing point 2 in the material model without failure criteria, the stresses would continue to increase with the strain increase towards point 5 and further. If the loading forces disappear in point 2, then the elastic forces will reduce the deformation to point 7 along path 2~7 and parallel to path 0~1. In the model with failure, point 5 corresponds to point 3 on the curve 2~4, where strength loss (softening) occurs. The 2~4 curve is called the degradation or failure curve defined by the parameter d, which is the damage evolution coefficient taking values from 0 to 1. The stress on the degradation curve is appropriate.
(18)$$\sigma =\left(1-d\right)\bar{\sigma }.\;$$

The material fracture occurs in point 4 after reaching the value of the fracture deformation $\varepsilon_{failure}^{pl}$ However, if during the degradation of the material on the curve 2–4 the element breaks or the forces loading the element disappear, e.g., in point 3, then the remaining elastic forces will reduce its deformation to point 6 along the 3–6 path, which is not parallel to the 0–1 path. The evolution of failure determines the degree of degradation at which failure of the material will occur. The value of *d* = 0 means that the plastic stress has reached the value of *R*_m,_ but the material has not yet been degraded, while the value of *d* = 1 means the complete degradation of the material. The failure evolution is described as a function of the plastic displacement of the *u*_pl_, defined as follows [13]: (19)$$u_{\mathrm{pl}}=L\cdot \varepsilon_{\mathrm{pl}}$$
where *L* is the characteristic length of the FEM element.

The rate of evolution of failure describes the path along which material degradation develops. In CAE programs, linear, exponential, and tabular descriptions are adopted. The linear relationship is expressed as the ratio of plastic displacement to failure displacement [13].
(20)$$d=\frac{u_{\mathrm{pl}}}{u_{\mathrm{failure}}}$$

Table 5 lists the points from the diagram in Figure 9, based on which the failure parameters for tensile strength of HY 80 steel were determined. 

Following these parameters, calculations were carried out for uniaxial stretching as follows:$$\varepsilon_{failure}=\varepsilon_{4}-\varepsilon_{7}=0.2280-0.0991=0.1289.\;$$
$$d\bar{\sigma }=\sigma_{5}-\sigma_{3}=836.0-489.9=346.1.\;\mathrm{MPa}$$
$$\mathrm{since}\;\sigma =\left(1-d\right)\bar{\sigma }.\;\mathrm{so}\;d=1-\frac{\sigma }{\bar{\sigma }}=1-\frac{489.9}{836}=0.414.\;$$
E’=(1−d)E=(1−0.414)·211= 124 GPa
$$u_{failure}=0.1289\cdot L$$

Summarising the tested HY-80 steel can be described by the following equations:

Young modulus: E=211 GPa;

Johnson–Cook model:$$\sigma =\left(559+518\cdot \varepsilon^{0.379}\right)\left[1+0.0268\;\cdot \ln\left(\frac{\dot{\varepsilon }}{0.0001}\right)\right]\left[1-{\left(\frac{\theta -293.15\;}{1\;470}\right)}^{1.14}\right]$$

Failure parameters:$$d=0.414;\;\varepsilon_{failure}=0.1289;\;\eta_{Triax}=0.33$$

## 5. Conclusions

Johnson–Cook HY 80 steel characteristics and material model were developed based on the static and dynamic tensile tests on the rotary hammer. Tensile tests performed on a rotary hammer allowed us to determine the mechanical properties of steel in the range of deformation speed 0–2000 s^−1^. The knowledge of the behaviour of steel for increased deformation rates enables the simulation of fast-changing processes such as a collision, projectile fire, impact of a shock wave (pressure from the explosion) on the tested object, or modelling of submarine implosion. The obtained data should be verified by an appropriate simulation and experiment, which will be the subject of the subsequent study.

The results of the tests of HY 80 steel after 55 years of operation show that the several decades of exploitation of this material in challenging sea conditions did not adversely affect its mechanical properties. They are close to catalogue values. The yield point of this steel is catalogued at 80 KSI (552 MPa). From the tests performed, the yield point of *R*_e_ = 605.6 MPa (*R*_02_ = 444.5 MPa) was obtained, and the strength limit was *R_m_* = 782 MPa with a deformation of 0.1, which proves that good plastic and strength properties were maintained. 

By analysing the mechanical properties, it can be concluded that the ship’s hull made of this steel without significant corrosion and operational losses could be used for the next years. One should be aware that it is still subject to erosive wear, which changes the overall strength of the hull. That may have an impact on limiting the maximum operational depth of the submarine.

Tests with a rotary hammer showed an increase in the strength of the steel with a reduced deformation. Unfortunately, due to the dynamic nature of the test and the possibility of potential damage to the extensometers, it was not possible to measure the deformation during the trial. This problem will be solved in the future with the use of high-speed cameras.

Increasing the strain rate in the range of up to 2000 s^−1^ increases the strength of the tested steel to 1140 MPa. That is a typical phenomenon in high-quality steel.

The study determined the failure parameters for the uniaxial tensile case (*η* = 0.33). The compression/tensile diagram for steel is symmetrical, which allows for the assumed failure criterion also for *η* = −0.33. The obtained amount of material did not qualify for a greater number of tests in which the failure parameters could be determined for the remaining characteristic values of the triaxiality coefficient.

## Figures and Tables

**Figure 1 materials-14-04213-f001:**
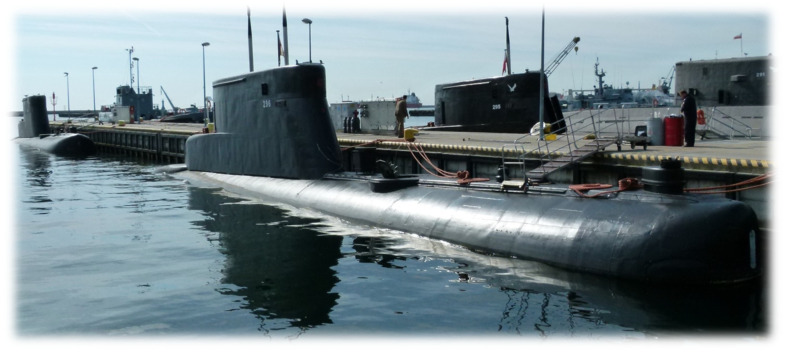
Submarines of project 207 (Kobben) in the home port in Gdynia [6].

**Figure 2 materials-14-04213-f002:**
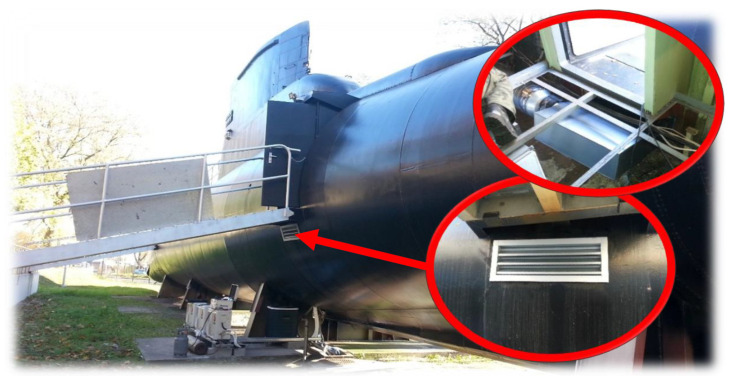
ORP “Jastrząb” (S-318 Kobben). Places from which material was taken for test samples.

**Figure 3 materials-14-04213-f003:**
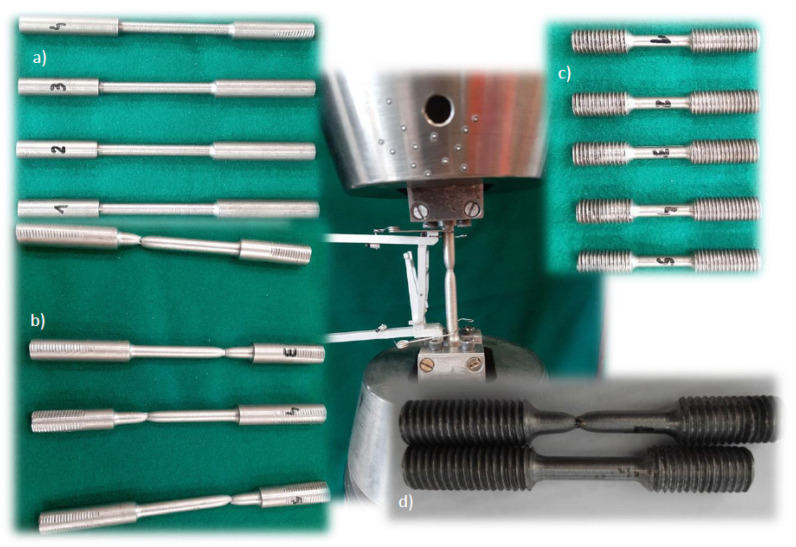
HY 80 steel specimens for static and dynamic tests: (**a**) samples for static tests on the MTS machine; (**b**) samples for static tests after fracture; (**c**) samples for dynamic tests on a rotary hammer; (**d**) samples for dynamic tests after fracture.

**Figure 4 materials-14-04213-f004:**
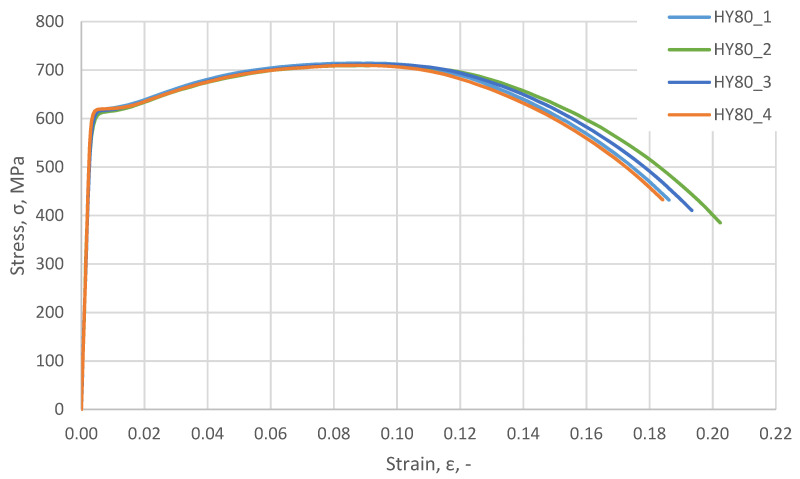
Tensile test results; nominal σ–*ε* of HY 80 steel, strain rate $\dot{\varepsilon }=0.0001\;s^{-1}$.

**Figure 5 materials-14-04213-f005:**
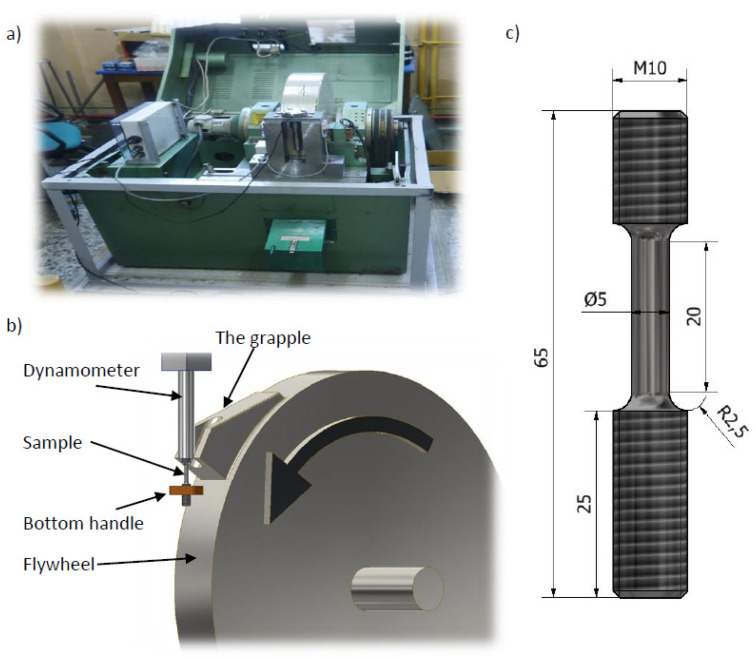
(**a**) Rotary hammer station (Fundamentals of Technology Laboratory, Polish Naval Academy), (**b**) scheme of dynamic tensile test on a rotary hammer, and (**c**) dimensions of a sample.

**Figure 6 materials-14-04213-f006:**
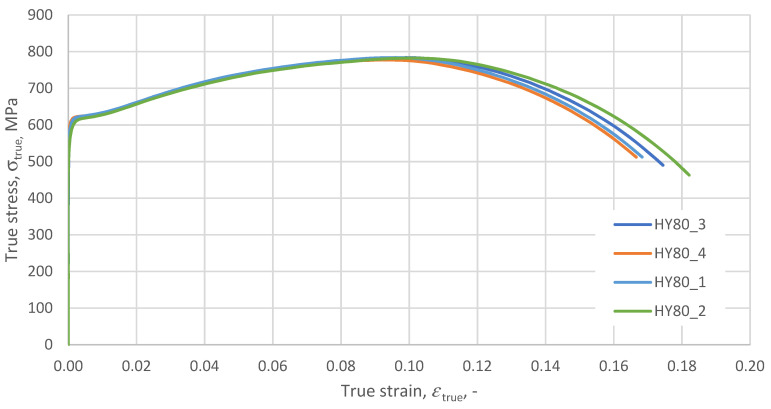
HY 80 σ_true_-*ε*_pl_ true characteristics, strain rate $\dot{\varepsilon }=0.0001\;s^{-1}$.

**Figure 7 materials-14-04213-f007:**
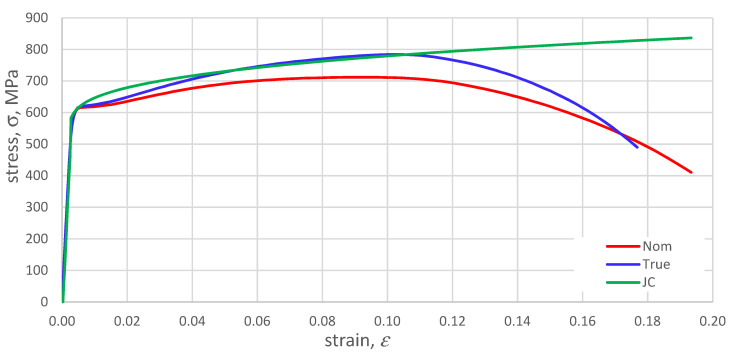
Comparison of the nominal, true characteristics and the proposed Johnson Cook HY 80 steel model, strain rate $\dot{\varepsilon }=0.0001\;s^{-1}$.

**Figure 8 materials-14-04213-f008:**
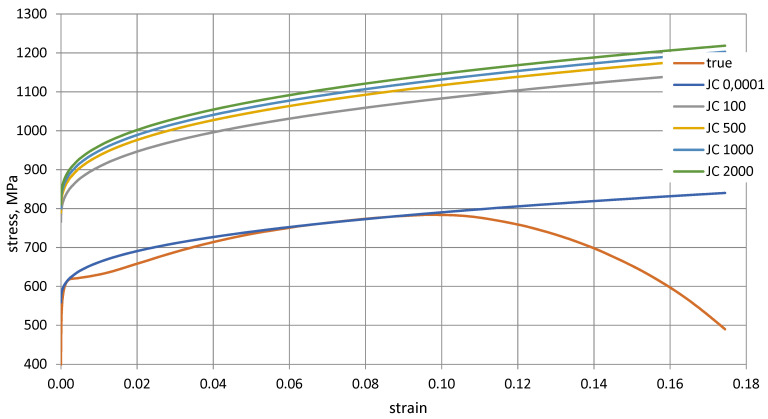
Johnson–Cook model of HY 80 steel as a function of strain rate, *C* = 0.0268.

**Figure 9 materials-14-04213-f009:**
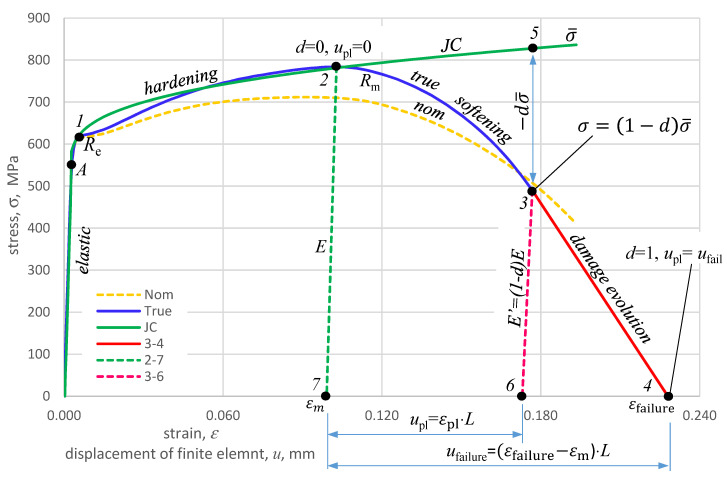
Failure diagram on the true characteristics of HY 80 steel (*σ*_true_-*ε*_true_).

**Table 1 materials-14-04213-t001:** Summary of test results on a rotary hammer.

Sample Name	*φ*	Measuring Length	Area *A*_0_	Breaking Force *F_m_*	Hammer Rotational Speed	Strain Rate	Dynamic Ultimate Strength Rm
	**mm**	**mm**	**mm^2^**	**kN**	**m/s**	**s^−1^**	**MPa**
HY 80_d1_v10	5.03	18.69	19.86	25.13	10.00	535	1265.28
HY 80_d2_v20	5.03	19.36	19.86	30.35	20.00	1033	1528.10
HY 80_d3_v30	5.02	19.33	19.78	30.76	30.00	1552	1554.92
HY 80_d4_v40	5.07	18.53	20.18	31.41	40.00	2159	1556.62

**Table 2 materials-14-04213-t002:** Material constants describing HY 80 steel based on Formulas (3)–(7).

Sample Name	Young Modulus	Yield Point	Yield Strain	Ultimate Strength	Ultimate Strain	Proof Load
	*E* GPa	*R_e_* MPa	*ε_e_* -	*R_m_* MPa	*ε_m_* -	*A = σ*_pl = 0_ MPa
HY 80_1	208.6	605.9	0.0041	783.9	0.1028	563.9
HY 80_2	210.8	610.5	0.0037	777.5	0.0958	576.0
HY 80_3	214.6	604.4	0.0037	784.1	0.0996	561.2
HY 80_4	210.7	601.7	0.0044	782.6	0.1045	536.0
**Average**	**211.2**	**605.6**	**0.0040**	**782.0**	**0.1007**	**559.3**

**Table 3 materials-14-04213-t003:** Ultimate strength for various strain rates.

$\mathrm{Strain}\;\mathrm{Rate},\;\dot{\varepsilon }$	${\dot{\varepsilon }}_{0}=0.0001s^{-1}$	$\dot{\varepsilon }=535s^{-1}$	$\dot{\varepsilon }=1033s^{-1}$	$\dot{\varepsilon }=1555s^{-1}$	$\dot{\varepsilon }=2159s^{-1}$
$R_{m,\left(\dot{\varepsilon }\right)}$ **, MPa**	782.00	1047	1115	1130	1140
**C**	-	0.021873	0.026366	0.026877	0.027108

**Table 4 materials-14-04213-t004:** *η*_TRIAX_ values for selected 3D cases [3].

**3D Cases**	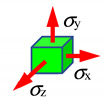	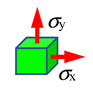	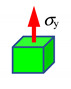	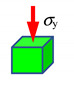	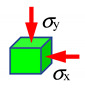	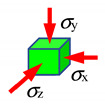
**Principal stresses**	*σ*_x_ = *σ*_y_ = *σ*_z_>0	*σ*_x_ = *σ*_y_*σ*_z_ = 0	*σ*_y_ > 0*σ*_x_ = *σ*_z_ = 0	*σ*_y_ < 0*σ*_x_ =*σ*_z_=0	−*σ*_x_ = −*σ*_y_*σ*_z_ = 0	*σ*_x_ = *σ*_y_ = *σ*_z_<0
***η*** **_TRIAX_**	∞	0.66	0.33	−0.33	−0.66	−∞

**Table 5 materials-14-04213-t005:** The values in Figure 9 used in the calculations.

Point Label	Strain	Stress	Remarks
	***ε*** **_el_** **, -**	***σ*** **_true_** **, MPa**	
1	0.0040	605.6	Yield point *R*_*e*_
2	0.1028	783.8	Ultimate tensile strength *R*_*m*_
3	0.1768	489.9	Sample fracture
4	0.2280	0.00	*d* = 1 material total degradation
5	0.1768	836.0	Stresses in the material model without failure parameters
6	0.1730	0.00	Fracture deformation
7	0.0991	0.00	Deformation at ultimate strength *R_m_*, *d* = 0

## Data Availability

Not applicable.

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
