# Peer review of "Material Properties of HY 80 Steel after 55 Years of Operation for FEM Applications"

_materials, 2021, doi:10.3390/ma14154213_

Round 1
Reviewer 1 Report
The authors conducted mechanical property characterization tests to establish the Johnson-Cook relationship for a submarine steel in 55 years of service. The information could be of interest to engineers concerning the long-term mechanical performance of such steels.
There are some minor points that authors can clarify for publication:
p2-line 50: 0.12÷02% carbon. "÷" is not a symbol conventionally meaning "from ... to ...", use a different symbol such as "~". Also, does 02 mean 0.2?
in Fig. 5, 7 and 8, indicate the test strain rate.
p12-line229, "Tensile tests" instead of "Tests"; line 251, "high-speed camera" instead of "superfast camera".
Author Response
please see the PDF file

Reviewer 2 Report
In this manuscript, tensile specimens were made from the exterior surface of a submarine (S-318 Kobben) used by the Polish navy for about 55 years and tensile tests were performed. The parameters for using the Johnson-Cook model were calculated through static, dynamic tensile tests.
The weakest point of this manuscript is the research motivation. The benefits of measuring the physical properties of HY 80 steel, which has been used for 55 years, are questionable. The authors emphasize that it can be used for finite element analysis, but there is little possibility of re-creating HY 80 steel used for submarines for 55 years and using it for other purposes. The reviewer recommends not publishing this article on ‘Materials’.
Other comments and suggestions are referred to in the attached file.

Author Response
please see the PDF file

Reviewer 3 Report
The paper regards interesting and important theme in the field of Materials Science and Engineering. The paper fits the Journal’s aims and scopes.
In particular, the authors address the question of the decades-long service of the considered steel on its key mechanical characteristics. The experimental part of the work is rather well designed, the methodology is adequate, the test execution, data acquisition and treatment are correct, and the results are satisfactorily presented. The paper provides useful and extensive reference data for the assessment of the residual operability of the widely used shipbuilding steel.
In my opinion, this is a kind of paper that may be published in the Materials journal.
Nevertheless, I would like to suggest some improvements, and also to ask the authors kindly to explain more extensively some issues that remain not clear for me. I guess, this could be beneficial for the readership.
First, I am bewildered by your strong focusing (from the very title of the paper) on the Finite Element Method (FEM) applications. But you study and characterize the material behavior, that is, the objective reality, which is nether dependent on nor associated with any method (numerical or not) of solution of any boundary value problems or equations. I guess, your material behavior description may be used to evaluate the stress-strain state and the behavior of, say, a cylindrical tube of a sphere under pressure, which can be done straightforwardly with a pencil and paper. I mean, your characterization of material is not a particular issue of FEM. I’d rather advice not to mention FEM at all in the paper.
Next, I would ask the authors to drop the stopping-words like “background”, “methods”, etc. in the abstract. They hamper smooth reading.
The sentence “The oldest in operational use” (line 41) has neither subject, not predicate. Please, correct.
The general requirement concerning the whole manuscript: please, define all variables and mathematical symbols at the first use. This is obligatory common practice in technical writing. Besides, please, do not use the same symbol for different things (like, e.g., A in eqs. (2)-(3) and in eq. (8). As well, all the abbreviations (like FEM, ORP, etc.) should be deciphered at the first use.
Regarding the transforming the nominal stress into the true stress, did you take into account the necking, which is visible in Figure 4? If not, your “true stress” hardly can be accepted as really true stress, but a lower-bound estimation.
You put eqs. (2) to (4) and then say “So: …” putting there eq. (5) as if this latter could be derived from the former equations. As far as I see, it cannot. Please, wherefrom the logarithmic strain arises.
In your result presentation, Figures 7 to 10 seem to have wrong denomination of the abscissa axis as “plastic strain epl”. I mean, plastic strain appears when the stress exceeds the elasticity limit value, which is not zero, but your curves imply, that plasticity appears immediately when loading starts – the curves start at the point (plastic strain = 0, stress = 0), whereas they should start at the point (plastic strain = 0, stress = elasticity limit > 0) with elasticity limit > 0. Didn’t you mean the total strain there? BTW, discussing the Figure 10 you speak about elastic regime between the points 1 and 2. On the “plastic strain-stress” any elastic regime must not appear.
Regarding Figure 8, if the common definitions are “nominal stress = force per undeformed section area”, “ true stress = force per reduced deformed section area”, what is the meaning of “JC true stress”: force per what area?
The schemes of applied forces on cubes in Table 4 are dumbfounding. It would be better (and natural) to put there the forces that keep the cube in equilibrium.
Regarding Figure 10 and its discussion, as far as the plastic strain is there dealt with, the statement “If the loading forces disappear in point 2, then the elastic forces will reduce the deformation to point 7 along path 2 ÷ 7 parallel to the path 0 ÷ 1.” (lines 187-188) is false. Plastic deformations, by definition, are irreversible=permanent deformations that remain in a solid after unloading, an as such they cannot be reduced by any “elastic forces” (in addition, it is not known what are they when loading is when the load is taken off.
To terminate, in the conclusions section you declare your intentions about what you are you going to in the future. This is not any conclusion.
Author Response
please see the PDF file

Round 2
Reviewer 2 Report
The authors faithfully revised the paper according to the reviewers' comments.
In the 1st review, I recommended rejecting for lack of research motivation. Looking at the revised manuscript, I thought that the evaluation of the mechanical properties of a submarine used for 55 years was meaningful in itself. The reviewer recommends publishing the manuscript as it is.